# A Nonuniformity Correction Method Based on 1D Guided Filtering and Linear Fitting for High-Resolution Infrared Scan Images

Bohan Li [1,2], Weicong Chen [1,2] and Yong Zhang [2,*]

1   University of Chinese Academy of Sciences, Beijing 100049, China
2   Shanghai Institute of Technical Physics Chinese Academy of Sciences, Shanghai 200083, China
*   Correspondence: zhangyong@mail.sitp.ac.cn

**Abstract:** During imaging, each infrared focal plane linear array scan detector detection unit determines a row of pixels in the image output. This sensor's nonuniformity appears as horizontal stripes. Correcting nonuniformity in high-resolution images without destroying delicate details is challenging. In this paper, a single-frame-based nonuniformity correction algorithm is proposed. A portion of a single-frame picture is intercepted initially. The 1D column guided filter is applied to smooth the captured image in the vertical direction. Then, the smooth image and high-frequency component with horizontal stripes and texture information are obtained. The subsequent step is to use the smooth portion of the image as the guided image and the high-frequency portion of the image as the input, so that the estimated nonuniformity noise of the image may be extracted using a 1D row guided filter. The segment of the corrected image is then obtained by subtracting the estimated nonuniformity noise from the segment of the raw image. The correction coefficients could be obtained by performing a linear regression fit on the pre- and post-guided filtering image segments. With the correction coefficients, the entire image could be corrected. Based on qualitative and quantitative analysis, the proposed algorithm outperforms other current advanced algorithms in terms of nonuniformity correction and real-time performance.

**Keywords:** guided filtering; nonuniformity correction; infrared focal plan linear array detector; linear regression fitting

## 1. Introduction

In recent years, infrared imaging systems have been greatly improved in terms of technical conditions and manufacturing processes, and they are now widely used in the fields of thermal imaging, fire detection, and aerial small target reconnaissance. The common infrared detectors are divided into infrared focal plane array detectors and infrared focal plane linear array scan detectors. Nevertheless, when the infrared detector is imaging, the response coefficient of the detector will fluctuate correspondingly under the impact of variations in the ambient temperature and the voltage of the imaging circuit, resulting in fixed pattern noise in the infrared system imaging [1,2]. This type of noise is typically difficult to eliminate directly at the hardware level, and the most prevalent techniques for compensating for hardware faults fall into two categories: reference-based correction methods [3] and scene-based correction methods [4,5].

The reference-based correction methods generally require the acquisition of blackbody temperature images, which are later used for the corrections performed by calibration. Common methods include single-point correction [1,6], two-point correction [7,8], multi-point correction [9,10], and polynomial fit correction [11,12]. Prior to being put into service, infrared detectors are typically adjusted using this procedure. Changes in temperature, voltage, and other environmental conditions will affect the detector response coefficients when the infrared detector is in use. Currently, if the reference-based correction method

is continued, it is necessary to cease equipment use and recalibrate infrared detectors, which severely impairs the normal operation of infrared detectors. To remedy this issue, scene-based non-uniform correction methods use the textural qualities of the acquired scenes to correct the images in real time, allowing for the detector to be recalibrated without the need for the blackbody.

In 2012, Tender et al. [13] proposed a nonuniformity correction method for infrared focal plane array detectors based on histogram statistics. This method can correct the nonuniformity of the image by modifying the histogram of the adjacent column of the image of an infrared array. However, this algorithm requires an image histogram forward transform as well as an inverse transform, and it has a significant temporal complexity. He et al. [14] first proposed a guided image filtering method in 2012. Y. Cao et al. [15] applied the guided filtering method to the nonuniformity correction of the infrared focal plane array detector in 2016. This method effectively removed the nonuniformity noise of the array image by constructing a one-dimensional guided filtering window and conducting twice guided filtering operations. Guided filtering needs the calculation of linear coefficients for each window, making it more time-consuming for high-resolution images. In 2018, Y. Cao et al. [16] proposed a denoising method based on multi-level wavelet transforms and guided filtering. This method corrects the nonuniformity of the array image by conducting guided filtering operations on the vertical high-frequency components of the image after the wavelet transform. After the wavelet transforms, the guided filtering only targets the high-frequency image components, which drastically reduces the temporal complexity of this method. However, the approach processes the wavelet transform in the frequency domain and tends to blur the image details. After that, Ende Wang et al. [17,18] combined TV regularization, wavelet transform, and guided filtering to the nonuniformity correction of the focal plane array detector. Due to frequency domain denoising, all of these methods may blur visual features and produce ghost effects. In addition, the multi-frame nonuniformity correction methods [19–23] and nonuniformity correction methods [24–27] based on a deep learning model are also applied to the field of image nonuniformity correction.

All of the aforementioned techniques are nonuniformity correction techniques for infrared focal plane array detectors. In this paper, based on the detector's features, a linear rectification model of an infrared focal plane linear array scan image is proposed using guided filtering. In this paper, we first extract a portion of the infrared linear array scan image and then apply 1D column guided filtering operations to a blurred image and the high-frequency component of the original image. The high-frequency component image is taken as the input image, and the blurred image is taken as the guide image. The 1D row guided filtering is performed to extract the nonuniformity noise of the linear array scan image. By subtracting the estimated nonuniformity noise image from the original image, the partial linear scan array image corrected by the guided filter is obtained. Finally, using the linear regression model, the linear correction coefficients are calculated to complete the nonuniformity correction of the entire frame.

Compared to the other correction algorithms [13–27], the method in this paper has the following advantages. First of all, this approach employs only a portion of a linear array scan image to calculate the linear model's correction coefficients and complete the frame's correction. The image resolution used for guided filtering calculation is significantly lowered as well as the computational complexity, which is well suited to satisfy the real-time demands of high-resolution infrared focal plane linear array scan images. Secondly, since the same row of pixels in the infrared scan image is determined only by a single detection unit, the algorithm in this study combines the principles of infrared column detector calibration and guided filtering to develop a one-dimensional linear regression coefficient model, which further improves the image's correction effect. The proposed approach corrects the nonuniformity in the spatial domain without degrading the image's high-frequency texture features or introducing ghost artifacts. Thirdly, the approach presented in this paper is a single-frame image algorithm that does not require consideration of the influence of camera jitter and scene heat radiation variations on nonuniformity correction

and has high robustness. Finally, the approach presented in this research is superior to the correction algorithm based on deep learning that disregards, without considering the creation of the dataset.

## 2. Proposed Approach

Our proposed infrared image nonuniformity correction approach comprises two processing stages: (1) to intercept a portion of an entire visual frame and apply two one-dimensional guided filtering operations to it; (2) to use the portion of the image before and after guided filtering to calculate the linear correction coefficients and apply them to correct the whole frame image. The result of our proposed algorithm's non-uniform correction on the real infrared scanning image is depicted in Figure 1. The processing pipeline of guided filtering is depicted in Figure 2. The complete processing pipeline is schematically illustrated in Figure 3.

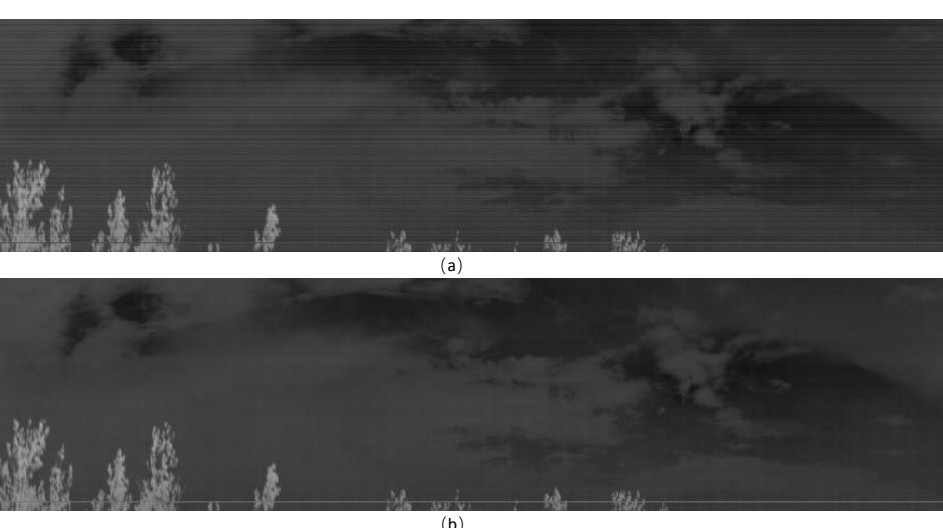

(a)

(b)

**Figure 1.** Non-uniform correction effect of real infrared scanning image. (**a**) Raw infrared scanning image. (**b**) Non-uniform correction results of our proposed algorithm.

### 2.1. Nonuniformity Correction Model Based on 1D Guided Image Filtering

The guided filter denoising model was first proposed by He [14] in 2012. In 2015, Cao [15] applied the guided filter denoising model to the nonuniformity correction of infrared uncooled array detectors. They obtained good results. In this subsection, we propose a novel non-uniform correction model based on two 1-D guided filtering for infrared focal plane linear array scan detectors. The implementation of guided filtering [14] is based on the assumption that the guided picture and the output image captured in a limited window have a linear connection, as follows:

$$q_i = a_k v_i + b_k, \forall i \in w_k \tag{1}$$

where $q_i$ is the pixel value of the output image, $(a_k, b_k)$ is the linear coefficients in the small window, and $v_i$ is the pixel value of the guided image. $w_k$ is the window, and noise characteristics must be taken into account when designing the window form. The guided filtering is completed by constraining the minimum difference between the output image $q_i$ and the input image $u_i$:

$$E(a_k, b_k) = \sum_{i \in w_k} ((a_k v_i + b_k - u_i)^2 + \varepsilon a_k) \tag{2}$$

where $E(a_k, b_k)$ is the cost function, $\varepsilon$ is a regularization parameter penalizing large $a_k$. The results of coefficients $(a_k, b_k)$ [14] is shown as

$$a_k = \frac{\frac{1}{|w|}\sum_{i\in w_k} v_i u_i - \mu_k \bar{u}_k}{\frac{1}{|w|}\sum_{i\in w_k} v_i^2 - \mu_k^2 + \varepsilon} \tag{3}$$

$$b_k = \bar{u}_k - a_k \mu_k \tag{4}$$

where $\mu_k$ is the mean of the guide image $v$ in the window, $\bar{u}_k$ is the mean of the input image $u_i$ in the window $w_k$, and $|w|$ is the number of pixels in the window $w_k$. $\varepsilon$ is a manually set parameter, which determines the strength of local averaging. Its effect is similar to the range variance parameter $\sigma_r^2$ in a bilateral filter [28]. However, a pixel $i$ is included in all overlapping windows $w_k$ that cover $i$, so the value of $q_i$ in (4) is not the same when computed in various windows. A basic technique is to calculate the mean of all conceivable values of $q_i$. Thus, after computing $(a_k, b_k)$ for each window $w_k$ in the image, we compute the output of the filtering by

$$q_i = \frac{1}{|w|}\sum_{k|i\in w_k} (a_k v_i + b_k) = \bar{a}_i v_i + \bar{b}_i \tag{5}$$

where $k|i \in w_k$ defines a number of overlapping windows $w_k$ covering the pixed $i$, $\bar{a}_i = \frac{1}{|w|}\sum_{k\in w_i} a_k$, and $\bar{b}_i = \frac{1}{|w|}\sum_{k\in w_i} b_k$ are the coefficient averages of all windows overlapping pixel $i$. For the nonuniformity of the infrared focal plane linear array scan detector, this paper designed a guided filtering correction model, as shown in Figure 2.

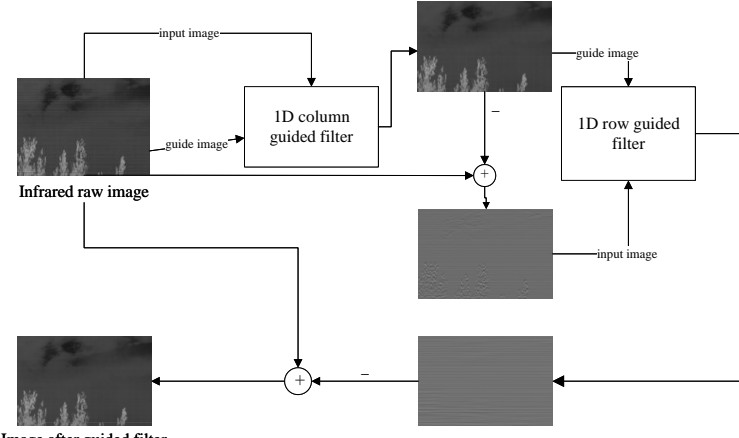

**Figure 2.** Guided filter processing workflow of the proposed method for horizontal stripe.

Since the nonuniformity of the focal plane linear array scan image appears as horizontal strips, the algorithm first takes the infrared raw image $I_p$ as a guided image and an input image. It smooths the image through a 1D column guided filter window, and at this point, Equations (3) and (4) become Equations (6) and (7).

$$a_k' = \frac{\sigma_k^2}{\sigma_k^2 + \varepsilon} \tag{6}$$

$$b_k = (1 - a_k')\mu_k \tag{7}$$

where $\mu_k$ and $\sigma_k^2$ are the mean and variance of the image $I_p$ in the window $w_k$. Equation (1) is used to conduct guided filtering operations, and the output image $I_u$ is the smoothed image in the vertical direction. An image $I_n$ containing nonuniformity noise and part of the image's high-frequency texture is obtained by subtracting $I_u$ from $I_p$, and the formula is given by

$$I_n = I_p - I_u \tag{8}$$

In order to better extract the nonuniformity noise from the image $I_n$, a 1D row guided filtering window is designed in this paper, with $I_u$ as the guided image and $I_n$ as the input image. The output image $I_s$ is obtained by guided filtering using Equation (5). We assume that the image $I_s$ contains most of the nonuniformity noise and little horizontal high-frequency texture detail. The image after nonuniformity correction is obtained by subtracting $I_s$ from $I_p$, as follows:

$$I_q = I_p - I_s \tag{9}$$

### 2.2. Linear Fitting Correction Model Based on Guided Filtering

Nonuniformity has been effectively corrected by the guided filter model. However, the image's high-frequency information may be unnecessarily blurred by the process of guided filtering. Since the infrared focal plane linear array scan detector imaging speed is fast, there would be tens of thousands of columns of images per second, and the guided filter needs to calculate the linear coefficients $(a_k, b_k)$ of each window, as shown in Equations (3) and (4). The algorithm has a high time complexity and large computational overhead, making it difficult to ensure real-time performance. Traditionally, the nonuniformity correction methods of an infrared array detector involve collecting the temperature of a blackbody and employing a linear model with two or more points to adjust. This is consistent with the assumption that the guided image and the output image are linearly related in the window. Therefore, the small window of guided filtering is discarded. The guided filtering window is a one-dimensional horizontal window, and its length is equal to the width of the guided image. The linearity coefficients are calculated using the image $I_q$ after the guided filtering as the input image and the raw infrared image $I_p$ as the guided image [29]. At this moment, the coefficients of the guided filter only need to be calculated once, and the computation of the linear coefficients is turned into least squares linear regression, which significantly decreases the algorithm's temporal complexity, as follows:

$$a(i) = \frac{\sum\limits_{j=1}^{W} I_q(i,j)\left(I_p(i,j) - \sum\limits_{k=1}^{W} I_p(i,k)/W\right)}{\sum\limits_{j=1}^{W}\left(I_p(i,j) - \sum\limits_{k=1}^{W} I_p(i,k)/W\right)^2} \tag{10}$$

$$b(i) = \frac{\sum\limits_{j=1}^{W} I_q(i,j) - a(i)\sum\limits_{j=1}^{W} I_p(i,j)}{W} \tag{11}$$

where $W$ is the width of the image after a guided filtering process, $I_p(i,j)$ is the image that was partially extracted from the full-frame raw infrared image with a horizontal strip, and $I_q(i,j)$ is $I_p(i,j)$ after guided filtering, as shown in Figure 3. The obtained linear correction coefficients are used to correct the whole frame of the raw infrared image:

$$\hat{I}(i,j) = a(i) \times \hat{I}_p(i,j) + b(i) \tag{12}$$

where $a(i), b(i)$ are the correction coefficients of the $i$ th row of the image, $\hat{I}_p(i,j)$ is the whole frame raw infrared image, and $\hat{I}(i,j)$ is the whole frame image after nonuniformity correction.

### 2.3. Proposed Workflow of the Proposed Method for Strip Nonuniformity Correction

The overall flow chart of the proposed algorithm in this paper is shown in Figure 3. There are specific steps in the algorithm.

1. A portion of the raw infrared scan image is intercepted and used to calculate the linear correction coefficients. After the experiments (see Section 3.2), the linear correction coefficients are calculated using the 1500 image columns chosen for this work.

2.  The selected image is used to eliminate the horizontal strip caused by nonuniformity with the guided filtering model proposed in Section 2.1.
3.  We use the linear fitting model based on guided filtering proposed in Section 2.2 to calculate the linear correction coefficients $a(i), b(i)$.
4.  The whole frame of the infrared image is corrected using linear coefficients as shown in the Formula (12).

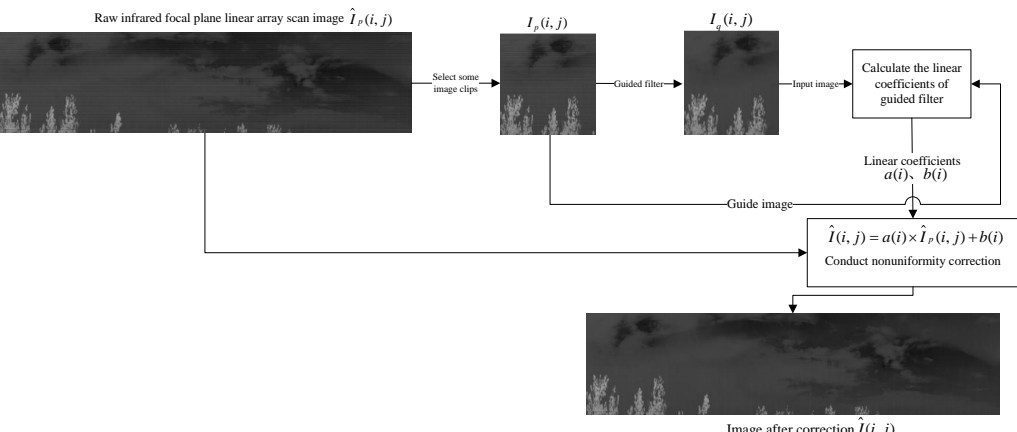

**Figure 3.** The overall flow chart of the proposed algorithm for nonuniformity correction.

## 3. Experiments and Comparison

This paper compares this algorithm against the most recent nonuniformity correction algorithms for infrared detectors to validate the algorithm's correction capabilities. The nonuniformity correction algorithm based on multi-level wavelet transform and guided filtering was proposed by Cao [16] in 2018. The nonuniformity correction algorithm based on wavelet decomposition and TV regularization was proposed by Wang [18] in 2020, and the other is the algorithm based on wavelet decomposition and image column equalization proposed by Wang [17] in 2019. The data in this paper are images from the FLIRADAS dataset, which includes 4224 infrared images, each with a resolution of 480 × 640. By analyzing the noise of the focal plane linear array, the simulated horizontal stripe is added to the image of the FLIRADAS dataset in order to quantitatively and qualitatively evaluate the algorithm's correction effect. The infrared long-wave scan detector acquires the real image set of the long-wave infrared scan, with a total of 50 frames and a resolution of 3053 × 55,000. This research uses 20 frames of scan images without nonuniformity to design the algorithm's parameters. For qualitative examination, 30 frames of real scan images with nonuniformity are used.

### 3.1. Noise Modeling Analysis

The causes of nonuniformity in infrared images are very complicated. From the perspective of signal transmission, the nonuniformity of sensors is mainly caused by two reasons. One is the inconsistent response of the photosensitive element of the infrared detection unit, while the other is the impact of the readout circuit's noise. According to the analysis in Song [3], the detector response model can be approximated as a linear model when the infrared detector operates normally, as indicated in the Formula (13):

$$Y(i) = g(i)R(i) + o(i) + \sigma(i) \tag{13}$$

where $Y(i)$ denotes the response value output by the $i$-th unit in the direction of the detector column. $R(i)$ indicates the actual amount of infrared radiation received by the detection device. $g(i)$ and $o(i)$ are the gain coefficient and bias coefficient of the infrared detection device, respectively. $\sigma(i)$ represents random noise. Random noise $\sigma(i)$ can be ignored when the signal-to-noise ratio of the infrared detector is high. For infrared scanning detectors, the output values of the same row of images have the same gain

and bias coefficients, so the nonuniformity of the detector appears as visible horizontal stripes. In conventional infrared detector nonuniformity correction, the black body is typically employed for image acquisition first, followed by the linear model for two-point correction [3], as illustrated by Equation (14). The objective is to ensure that each detector unit has an identical response curve.

$$\hat{I}(i, j) = k(i) \times v(i, j) + m(i) \tag{14}$$

where $\hat{I}(i, j)$ represents the image after correction, $k(i)$ represents the gain correction coefficient of the linear model, and $m(i)$ represents the bias correction coefficient. Analyzing the linear correction model of nonuniformity noise, this paper proposes an analog nonuniformity model of infrared detectors, as demonstrated by Equation (15).

$$I_{1n}(i, j) = g(i) \times I_1(i, j) + b(i) \tag{15}$$

where $I_1(i, j)$ is the infrared image without nonuniformity, $I_{1n}(i, j)$ is the image with added analog horizontal stripes, $i \in [1, H]$. $H$ is the image's height. $g(i)$ is a sequence of Gaussian distribution with a mean value of 1, and $b(i)$ is a sequence of Gaussian distribution with a mean value of 0. The horizontal stripe of an image is more apparent as the variance of the Gaussian sequence increases.

### 3.2. Image Correction Effect of Image Size Used in Linear Fitting Model

The two-step approach for infrared scan images is proposed in this research. First, a portion of the high-resolution scan image is intercepted, and then the intercepted image is corrected using a one-dimensional guided filter model. The next step is to calculate the coefficients of the linear fitting model and then use the correction coefficients to correct the full image frame. This section primarily studies the effect of intercepted image size on the correction. The experimental data consists of 20 frames of real infrared scanning images without nonuniformity to which horizontal stripe was added using Formula (15), where parameter $g(i)$ follows a Gaussian distribution with a mean of 1 and a variance of 0.02 and parameter $b(i)$ follows a Gaussian distribution with a mean of 0 and a variance of 0.02. The scan images with various column counts are intercepted, and the PSNR of the images after correction is computed. Table 1 displays the average findings of the full dataset.

**Table 1.** Image correction effect of image size used in linear fitting model.

| Images\Colums | 500 | 1000 | 1500 | 2000 | 2500 |
|---|---|---|---|---|---|
| **PSNR** | 39.79 | 40.33 | 40.76 | 40.78 | 40.74 |

Table 1 demonstrates that when 1500 scan image columns are picked for the calculation of correction coefficients, the image can already reach a good level. PSNR increases somewhat when additional scan image columns are selected to calculate correction coefficients. As the number of image columns increases, the PSNR of the corrected image tends to stabilize and does not grow with the number of image columns. This demonstrates that this approach is practical for the nonuniformity correction of the infrared scan image and can only use a portion of the image to calculate the correction coefficients needed to finish the correction of the entire image frame.

### 3.3. Image Correction Effect of Noise

The previous section's experiment demonstrated that when the number of intercepted image columns is 1500, the effect of the picture after correction becomes more effective and stable. The purpose of this section's experiment is to investigate the effect of noise on image correction. This research adds Gaussian white noise with various variances to the intercepted image. The diagram of the experimental example is presented in Figure 4.

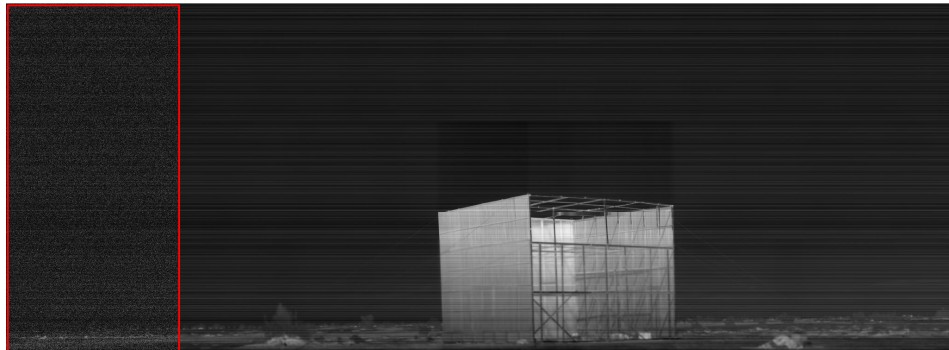

**Figure 4.** Infrared scan image with the addition of Gaussian noise. Due to the very high resolution of the infrared column scan image (3053 × 55,000), only a portion of the scanned image (3053 × 8192) is displayed in this figure. The image within the red border is the image utilized for the linear fitting model in Section 2.1, with a resolution of 3053 × 1500 pixels.

We add Gaussian white noise with varying variances to this portion of the image, and then use the Formula (15) to add horizontal stripes to the image. $g(i)$ conforms to a Gaussian distribution with a mean of 1, and $b(i)$ conforms to a Gaussian distribution with a mean of 0. $g(i)$ and $b(i)$ have the same variances which are shown in the first row of Table 2. The experimental data consist of 20 frames of infrared scan images without nonuniformity; the experimental outcomes are presented in Table 2.

**Table 2.** Influence of Gaussian noise in the intercepted image on correction algorithm.

| Variance | 25 | 36 | 49 | 64 | 81 | 100 | 121 | 144 | 169 | 196 | 225 |
|---|---|---|---|---|---|---|---|---|---|---|---|
| **PSNR** | 43.90 | 44.02 | 44.14 | 44.12 | 44.17 | 44.12 | 43.79 | 43.93 | 44.03 | 43.55 | 43.41 |

The image of the infrared column scan is an 8-bit image. The first row of Table 2 is the variance of the added Gaussian noise with a mean of 0 and the variance ranges from 25 to 255. The noise intensity increases as the variance of the Gaussian distribution increases. The second row is the average PSNR of the dataset following correcting. Table 2 demonstrates that when the variance of the added Gaussian noise grows, the PSNR of the image collection swings within a given range, and the connection is not linear. Consequently, the Gaussian noise has little effect on the denoising effect of the algorithm described in this paper.

*3.4. Qualitative Analysis of the Correction Effect of the Algorithms*

On the dataset, the algorithm proposed in this research is compared to the nonuniform correction algorithms Cao [16], Wang [17], and Wang [18] developed in recent years, and their correction effects are evaluated qualitatively. The data collection consists of five infrared images from the FLIRADAS dataset and two frames of real infrared scan images with nonuniformity. Figure 5 depicts the five raw infrared images picked from the FLIRADAS dataset. After adding stripes, the images are depicted in Figure 6. The noise parameter $g(i)$ in Formula (15) follows a Gaussian distribution with a mean value of 1 and a variance of 0.02, while $b(i)$ follows a Gaussian distribution with a mean value of 0 and a variance of 0.02. The resolution of the image in Figure 6 is relatively high, and when the image is magnified, the stripes of nonuniformity can be seen.

Since the comparative algorithms Cao [16], Wang [17], and Wang [18] all used wavelet transform in the process, different wavelet bases have a substantial impact on the image correction effect. By comparing the effects of each algorithm on different wavelet bases, the one with the best effect is selected for nonuniformity correction, and the results are compared to the algorithm presented in this paper. The common wavelet-based haar wavelet, sym8 wavelet, and db4 wavelet are used for comparison. The nonuniformity

correction effects of Cao [16], Wang [17], and Wang [18] with respect to various wavelet bases, are shown in Figure 7.

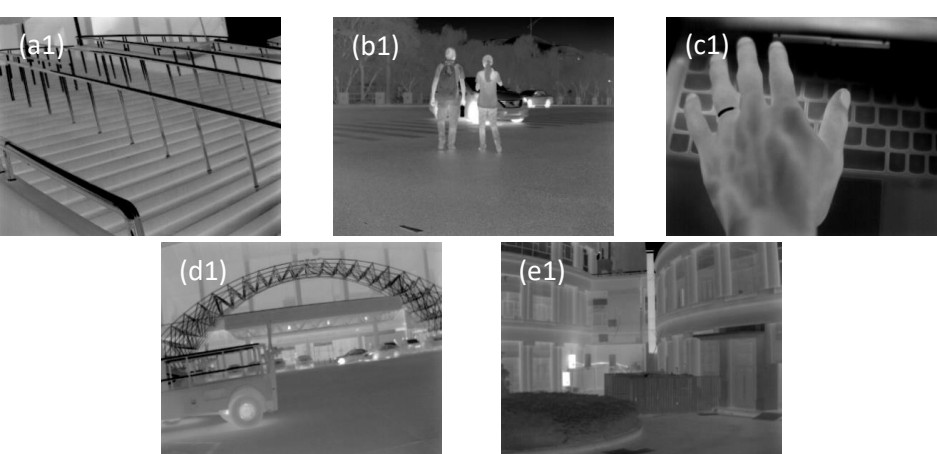

**Figure 5.** Original image (**a1**) stairs (**b1**) people (**c1**) hands (**d1**) bus (**e1**) buildings.

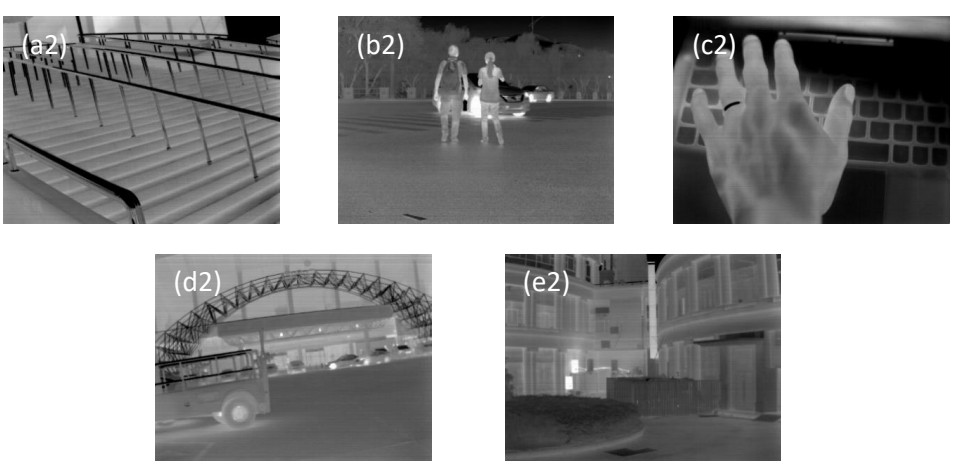

**Figure 6.** Image after adding analog stripes (**a2**) stairs (**b2**) people (**c2**) hands (**d2**) bus (**e2**) buildings.

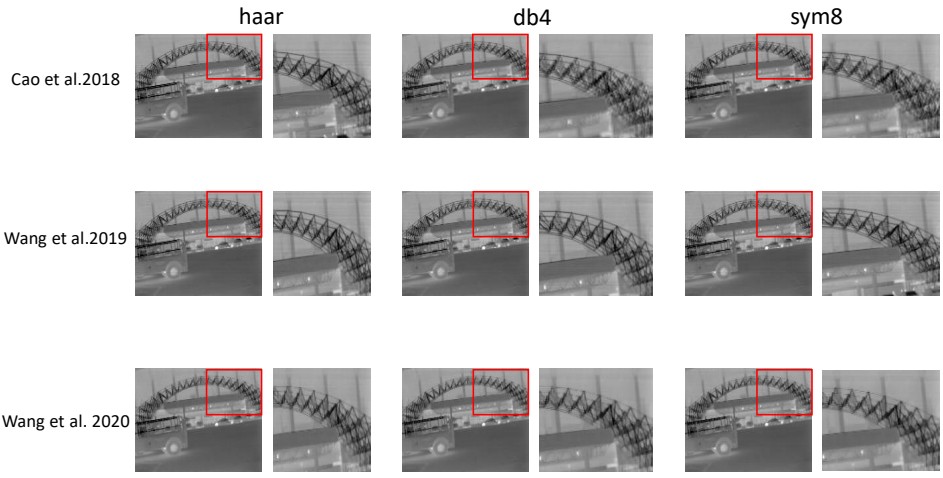

**Figure 7.** Nonuniformity correction effect of Cao [16], Wang [17], and Wang [18] with respect to distinct wavelet bases. The wavelet bases include "haar", "db4", and "sym8" wavelets. The image "bus" is added with stimulated nonuniformity noise. In order to highlight the correction effect of the algorithm on horizontal stripes and the fuzzy degree of the algorithm on image texture details, the images within the red border are enlarged.

Wang [17] has a good correction impact on the "db4" wavelet, and the texture of the image's details is less blurry. Wang [18] and Cao [16] have severe distortion of image texture detail with the "db4" and "sym8" wavelets, so they selected the "haar" wavelet. The size of the guided filter windows of Cao [16], Wang [17], and Wang [18] are all one-fourth of the width of the processed image sections. Since the simulated image is small, the algorithm in this paper selects the whole frame to calculate the linear correction coefficients without intercepting the image, where the size of the smoothing guide filter window is $8 \times 1$, and the size of the horizontal stripe extraction guide filter window is $1 \times 10$. The guide filter parameters $\varepsilon$ are all 0.16. Figure 8 depicts the effect of image nonuniformity correction using different algorithms.

It can be seen that Cao [16], Wang [17], and Wang [18] have certain effects on horizontal stripe, but they all blur some texture details of the image. Wang [18] has the best nonuniformity correction result on the smooth image, but the blurriest image texture details. Wang [17] and Cao [16] have an average effect on horizontal strip removal, although their harm to picture detail texture is not as severe as Wang's [18]. The damage to image texture details caused by the algorithms of Cao [16], Wang [17], and Wang [18] is mainly due to the fact that they all process the image's horizontal high-frequency information after wavelet transform, and the algorithms eliminate the stripes while also destroying the part of the image with the high-frequency information. In contrast, the algorithm presented in this paper linearly corrects the image in the spatial domain and does not process the image in the frequency domain. Consequently, it destroys fewer image texture details and outperforms the algorithms of Cao [16], Wang [17], and Wang [18] in terms of the nonuniformity correction effect. Finally, in order to better verify the nonuniformity correction effect of this algorithm, real infrared images containing non-uniform strips generated by long-wave infrared focal plane linear array scan detectors are used for experiments to compare the effect of each algorithm. Since the number of columns of a frame of infrared focal plane linear array scan image is about 50,000, in order to show the image's denoising effect, this paper intercepts the scanning image with an image resolution of $3053 \times 4096$. The algorithm in this paper uses 1500 columns of scan images to calculate the linear correction coefficients through guided filtering. The size of the smoothing guided filtering window is $12 \times 1$. The size of the guided filtering window for horizontal stripe extraction is $1 \times 100$, and guided filtering parameters $\varepsilon$ are 0.16. The algorithms of Cao [16], Wang [17], and Wang [18] all correct the $3053 \times 4096$ images directly.

The real scan image in Figure 9 shows that the scene is the sky, and the horizontal stripes caused by nonuniformity are plainly visible in the clouds. The effects of the Cao [16], Wang [17], and Wang [18] algorithms are not satisfactory in the high-resolution scan image, which contains a large number of horizontal stripes. The correction impact of the proposed algorithm is clearly superior to all others, and the horizontal stripes in the high-resolution cloud image are effectively erased.

*3.5. Quantitative Analysis of the Nonuniformity Correction Effect of the Algorithms*

In order to better analyze the image correction effect, this paper uses PSNR, roughness, and the energy of vertical gradient for quantitative analysis. In this quantitative analysis, the infrared image in Figures 5 and 6 with added simulated horizontal stripes and 100 randomly selected frames from the FLIRADAS dataset are employed. All images are 8-bit images. The definition of PSNR is described as follows:

$$\mathrm{PSNR} = 10\log_{10}\frac{255^2}{\mathrm{MSE}} \tag{16}$$

where MSE is the mean square error between the original image and the input image. The higher the value of PSNR means the image is closer to the original image, and the better the correction effect. The PSNR results are shown in Table 3.

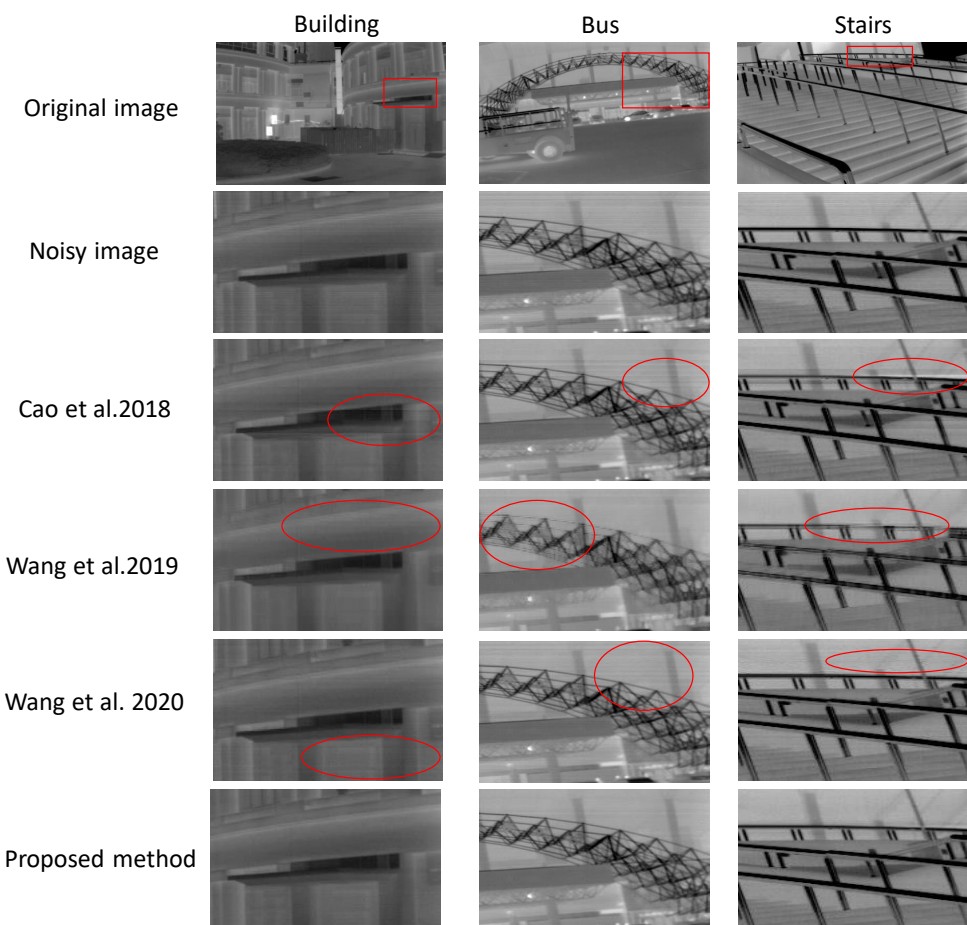

**Figure 8.** Nonuniformity correction effect of Cao [16], Wang [17], Wang [18], and our proposed algorithm. In order to better show the nonuniformity correction effect of each algorithm, this figure selects local magnified images of building, bus, and stairs to compare the effect of each algorithm. The parts in the red circle in the images are the parts where the non-uniform correction results of the contrast algorithm are not good.

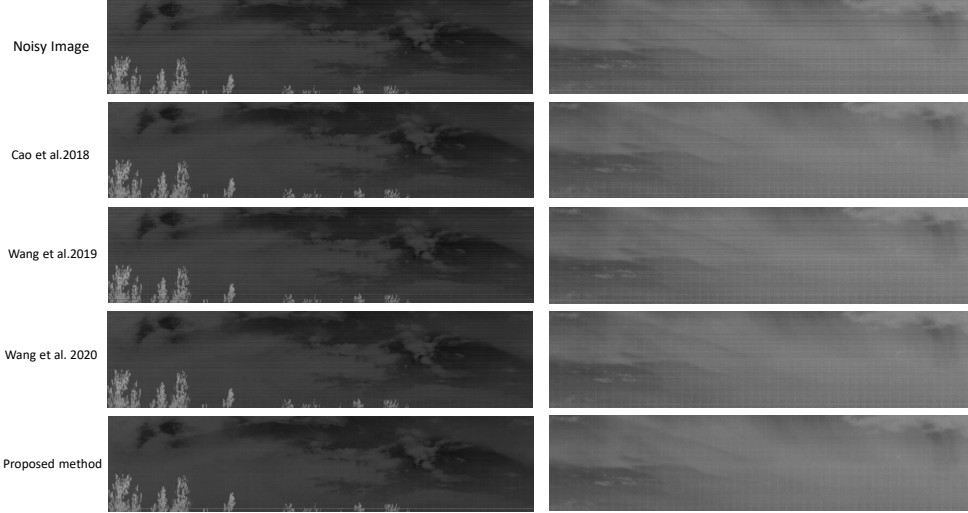

**Figure 9.** Real infrared focal plane linear array non-uniform correction of Cao [16], Wang [17], Wang [18], and our proposed algorithm. The resolution of the presented image is 1024 × 4096 due to the high resolution of the scanned image, as shown in this figure, in order to more accurately compare the denoising effect.

**Table 3.** PSNR results of Cao [16], Wang [17], Wang [18], and our method.

| Sequence\Method | Cao [16] | Wang [17] | Wang [18] | Proposed Method |
|---|---|---|---|---|
| bus | 40.16 | 41.18 | 31.90 | **46.24** |
| people | 40.80 | 42.72 | 36.19 | **45.30** |
| building | 42.35 | 43.48 | 39.17 | **47.20** |
| hand | 38.49 | 44.48 | 43.21 | **46.65** |
| stairs | 34.78 | 38.19 | 29.77 | **38.98** |
| dataset | 39.85 | 42.84 | 34.79 | **45.74** |

It could be seen that our proposed algorithm has the best nonuniformity correction effect with the highest PSNR value. Due to the TV regularization of the wavelet transform DC component, the technique of Wang [18] yields the lowest PSNR of the image. This means that the DC component of the whole frame is changed. The algorithm of Cao [16] has a lower image PSNR in the image "hand", which may come from the fact that this image has more high-frequency details; after the three-stage wavelet transform high-frequency component processing, the image obtains more distortion. The roughness is another indicator to measure the image correction effect, and its definition is as follows:

$$p = \frac{||h * g|| + ||h^T * I||}{||I||} \quad (17)$$

where $|| \cdot ||$ denotes the L-1 norm, $*$ means the convolution operation, $h = [-1, 1]$, and $I$ denotes the input image. The smoother the image means the lower the roughness of the image. However, this does not imply a superior picture correction effect. The process of smoothing the image may lose the image's high-frequency information. Therefore, this experiment considers that the closer the roughness of the image is to the original image, the better the effect is. The results of roughness are shown in Table 4.

**Table 4.** Roughness of Cao [16], Wang [17], Wang [18], and our method.

| Sequence\Method | Original | Cao [16] | Wang [17] | Wang [18] | Proposed Method |
|---|---|---|---|---|---|
| bus | **0.0741** | 0.0765 | 0.0787 | 0.0652 | **0.0740** |
| people | **0.0942** | 0.0962 | 0.0975 | 0.0863 | **0.0943** |
| building | **0.0621** | 0.0646 | 0.0662 | 0.0591 | **0.0623** |
| hand | **0.0877** | 0.0939 | 0.0936 | 0.0868 | **0.0891** |
| stairs | **0.0977** | 0.1009 | 0.1045 | 0.0939 | **0.0984** |
| dataset | **0.0834** | 0.0851 | 0.0876 | 0.0786 | **0.0841** |

According to the data in Table 4, the roughness of the corrected image generated by our proposed technique is the closest to that of the original image. The image roughness of Wang [18] is the lowest, and the image is the smoothest, which is consistent with the correction effects of different algorithms in Figure 8. Since the nonuniformity noise of the infrared column scan image is in the horizontal direction, the image is not as smooth as the original image in the vertical direction; the energy of vertical gradient is another indicator of the nonuniformity correction effect, and its definition is as follows:

$$E = \sum_{i=1}^{H-1} \sum_{j=1}^{W} \frac{[I(i,j) - I(i+1,j)]^2}{(H-1) \times W} \quad (18)$$

where $I(i,j)$ is the input image, $H$ is the height of the image, and $W$ is the width of the image. The results of the energy of vertical gradient are shown in Table 5.

**Table 5.** Energy of vertical gradient of Cao [16], Wang [17], Wang [18], and our method.

| Sequence\Method | Original | Cao [16] | Wang [17] | Wang [18] | Proposed Method |
|---|---|---|---|---|---|
| bus | **90.00** | 93.19 | 92.56 | 53.86 | **87.97** |
| people | **53.92** | 54.66 | 55.48 | 37.20 | **52.35** |
| building | **21.62** | 23.85 | 24.50 | 15.81 | **20.95** |
| hand | **23.50** | 32.56 | 27.12 | 21.74 | **23.45** |
| stairs | 173.16 | 168.90 | **176.44** | 116.31 | 162.74 |
| dataset | **75.98** | 71.93 | 78.29 | 50.43 | **72.62** |

According to Table 5, the algorithm in this paper has the closest energy of vertical gradient of the corrected image to the original infrared image in most cases. The vertical gradient energy of the corrected image produced by the Wang [18] method is the lowest and the images are smoother. This is consistent with the image in Figure 8.

From Figure 10, we can see that the proposed algorithm has the highest PSNR value of the corrected images, which indicates the best effect. It is worth noting that the PSNR of the corrected image of Wang's [18] algorithm with a variance $\sigma$ of 0.01 is smaller than that of the noisy image without denoising treatment. The main reason for this is that the algorithm applies TV regularization to the DC component of the image after the wavelet transform, causing the overall pixel value to alter. It can be seen from the effect in Figure 7 that the Wang [18] algorithm has the highest vertical smoothing degree for the image, and a better correction effect for horizontal stripes than the algorithms of Cao [16] and Wang [17]. Therefore, the PSNR of the corrected image varies slowly as the variance increases.

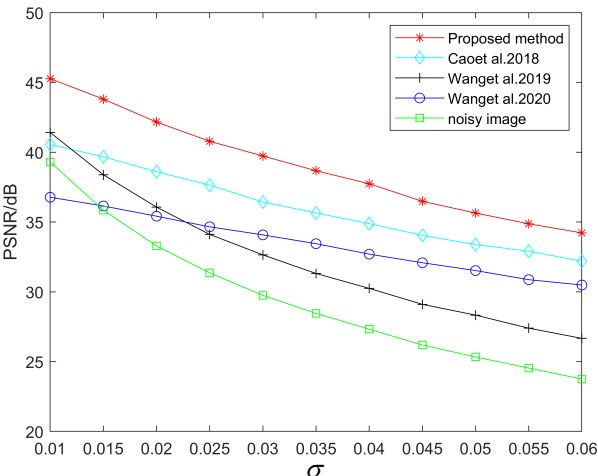

**Figure 10.** PSNR results of Cao [16], Wang [17], Wang [18], and our method. In order to verify the universality of the nonuniformity correction effect of the algorithm in this paper, 50 images are selected at random from the FLIRADAS dataset, and Gaussian cross-stripe noise with different variance $\sigma$ is added as shown in Equation (15). The mean PSNR values of the corrected images are calculated using various algorithms to create a relationship curve, as depicted in this figure.

### 3.6. Computational Time

Due to the fact that infrared focal plane linear array scan detectors image tens of thousands of columns per second, the algorithm must have high performance in real time. Cao [16] and other comparison algorithms perform the wavelet transform on the image. The size of a frame for an infrared focal plane array image is limited, and the image's length and width are around a few hundred pixels; thus, the wavelet transform is not a significant burden for it. For the infrared focal plane linear array scan image, the number of columns in a frame is tens of thousands, so the wavelet transforms applied directly to the whole frame of the line column image are not suitable, while the computational overhead is relatively large. In contrast, the proposed algorithm only requires a portion of the infrared

scan image to calculate the linear correction coefficients, which are then used to correct the image nonuniformity of the entire frame. The main cost of the algorithm is the computation of the linear correction coefficients across a portion of the image. Therefore, the proposed algorithm is less time-intensive and operates more in real time. To ensure a fair comparison, all techniques are executed in Matlab R2020b without optimizations or parallel-computing implementation on a PC with a 2.50 GHz Intel Core i7-9750HK processor and 16 GB of RAM. The experimental data in this section are partial images intercepted from real infrared focal plane linear array scan images with an image resolution of 3053 × 8192 and an A/D acquisition accuracy of 14-bit. In order to facilitate subjective evaluation, all 16-bit grayscale images are converted to 8-bit data grayscale images. The proposed algorithm in this paper selects the image with the resolution of 3053 × 1500 for calculating the linear correction model coefficients. The size of the vertical guided filter window for smoothing the image is 12 × 1, and the size of the horizontal guided filter for extracting the horizontal strip is 1 × 100. The artificial guided filter parameters $\varepsilon$ are both 0.16. Cao [16] adopts three-level image wavelet decomposition and haar wavelet. Wang [18] chooses the haar wavelet base and Wang's [17] algorithm chooses the "db4" wavelet. The experimental results are shown in Table 6.

**Table 6.** Computational time comparison of Cao [16], Wang [17], Wang [18], and our method.

| Algorithms | Cao [16] | Wang [18] | Wang [17] | Proposed Method |
|:---:|:---:|:---:|:---:|:---:|
| Time\s | 1.1014 | 120.2012 | 205.6192 | 1.0143 |

　　The algorithm execution time listed in Table 6 is the mean execution time of each algorithm after 10 iterations. An evident conclusion is that, when the image resolution is 3053 × 8192, the algorithm of Cao [16] and the algorithm proposed in this paper have similar minimum times of approximately 1s. In practical application, an infrared scan image frame contains tens of thousands of columns, and the device generates a frame of image in a short time of about 1–2 s. At this point, the processing time for a scanned image utilizing Cao's [16] algorithm will grow dramatically. By intercepting a portion of the image, the algorithm in this paper calculates the linear correction coefficients. The computational time of the proposed algorithm is mainly in the process of solving the linear model coefficients. The time required to generate an image is rather short. Since the infrared scan technology rapidly scans an image frame and the correction coefficients change very little, it can be considered that the correction coefficients do not change in a short time. Consequently, the production of an image frame often requires a simple linear correction. The time required to correct the infrared scan image will not increase significantly. The algorithm proposed by Wang [18] combines image wavelet decomposition with TV regularization. After wavelet decomposition, it takes 114.6317 s to correct the DC component of an image by TV regularization. This is mainly due to the fact that the DC component of the picture wavelet decomposition has a resolution of 1526 × 4096 and the image is huge; thus, the time required to correct it using TV regularization is longer. The algorithm proposed by Wang [17] combines the wavelet transform with the MHE method. It completes eliminating stripes by MHE equalization of the horizontal AC components of the image after wavelet decomposition and guided filtering. In the MHE algorithm, the adaptive selection of parameter s in the optimal Gaussian noise reduction operator $g_s(k) = \frac{1}{s\sqrt{2\pi}} e^{\frac{-k^2}{2s^2}}$ takes 184.4410 s, which is a relatively long time. To conclude, the algorithm of Wang [17] and Wang [18] have a high time complexity and are not suitable for nonuniformity correction of infrared focal plane linear array scan images with high resolution and demand for real-time performance.

## 4. Conclusions

　　In this paper, we propose a nonuniformity correction algorithm for an infrared focal plane linear array scan detector based on a single-frame image. Due to the high resolution of the infrared scan image, the algorithm for nonuniformity correction must have a high

real-time performance. In order to simultaneously fulfill the denoising effect and high real-time performance of the infrared sequence image, this paper presents a two-step correction approach. In the first stage, a portion of the line-sequence scan image is extracted from the whole frame scan image. On the intercepted picture, nonuniformity correction is accomplished using two one-dimensional guided filters. In the second step, since the corrected image may blur some high-frequency details, the algorithm in this paper calculates the linear correction coefficients using the portion of the image to reduce the damage to the image texture details caused by the guided filtering denoising algorithm. The linear correction coefficients are then used to correct the entire frame image. The proposed algorithm has the following advantages over other nonuniformity correction methods developed in recent years. Based on the real-time analysis of the algorithm, this study only uses a portion of the image to determine the correction coefficients for the nonuniformity correction of the entire frame. It meets the real-time needs of the scan images containing tens of thousands of columns per frame. The designed linear correction model conforms to the structural characteristics corresponding to the imaging of each row of the cycle scan image and the fixed detection unit, as determined by an analysis of the algorithmic principle. The algorithm provided in this paper is processed in the image space domain, does not involve image time-frequency conversion, and does not produce ghosting artifacts. The nonuniformity correction effect is superior to other algorithms. The technique in this paper is based on a single-frame image approach in terms of algorithm robustness. Unlike multi-frame denoising algorithms, the approach does not need to account for the influence of camera shake and scene heat radiation variations on the correction effect, and it is robust. In addition, unlike the conventional correction procedure that uses the black body to correct the detector's nonuniformity, the algorithm proposed in this paper does not require terminating the detector's use, allowing for the detector to be adjusted in real time during use.

**Author Contributions:** Methodology, B.L.; Formal analysis, B.L.; Resources, Y.Z.; Data curation, W.C.; Writing—original draft, B.L.; Writing—review & editing, B.L., W.C. and Y.Z. All authors have read and agreed to the published version of the manuscript.

**Funding:** This research received no external funding.

**Institutional Review Board Statement:** Not applicable.

**Informed Consent Statement:** Not applicable.

**Data Availability Statement:** The real infrared image dataset is generated by infrared longwave cooled linear scan detector.It is not a public dataset.The infrared focal array image data set comes from a public data set, https://camel.ece.gatech.edu/, accessed on 11 March 2023.

**Conflicts of Interest:** The authors declare no conflict of interest.

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
