# Peer review of "A Nonuniformity Correction Method Based on 1D Guided Filtering and Linear Fitting for High-Resolution Infrared Scan Images"

_applsci, doi:10.3390/app13063890_

Round 1

Reviewer 1 Report

 In this paper, the authors based on a qualitative and quantitative analysis, proposed an algorithm which outperforms other current advanced algorithms in terms of nonuniformity correction and real-time performance.

In general, the technical structure of the paper is well done. The idea although, it is a technical procedure is very well documented and can be considered novel.

The paper is well supported by mathematics and adequate comparisons are given with existing methods.

However,

Some recent references are needed.

The captions of the most figures must be more detailed, so that the figures are self-standing. This happens with tables as well.

The manuscript must be carefully revised as far as punctuation is concern.

The citations of the references in the text must be correct placed.

The authors must follow the formatting guidelines of the journal.

Additional comments

In subsection 2.1 the material from line 99 to 109 has to be improved. Equations 1, 2 and 3 are given without any explanations. The authors assume that the reader is familiar and aware of the mathematical reasoning. The simple exposition of the mathematical relations is not supporting for the propose method. Please improve subsection 2.1.

The same happens with the rest equations 4-7 which are coming as a result of the application of the Equation 1-3.

The authors should point out which is the novel contribution of theirs in the material of the subsection 2.1.

According to my opinion more analysis is required in subsection 3.1: Noise modeling analysis.

Author Response

Thanks for your constructive suggestion, which is valuable for improving the accuracy of the manuscript. I have done my best to revise my paper. Please read the ‘Revised_Version_Notes.docx’ first.

Reviewer 2 Report

The paper deals with an important practical task and the authors have proposed an interesting algorithm having quite high efficiency. However, I have several questions and propositions, namely: 

Line 39: what are “qualities” here?

What is meant by “robustness” in line 90?

Lines 93 and 94: two one- 93 dimensional guided filtering to it… Maybe, filtering operations?

It is worth showing one or two images at the beginning of Section 2 to demonstrate the effects against which You “combat”.

Line 99: it is unclear when He proposed his method in 2012 or in 2013 – look https://ieeexplore.ieee.org/document/6319316.

At the end of page 3: maybe has appeared or appears (not is appeared)

where after (9)

end of page 4: what is “is the correspond”?

again where after (10)

2.3 Proposed…

Correct Where to where after formulas in all places

I guess that PSNR is in the lowest line of Table 1. Please indicate this in text or somewhere else.

Variance values in Table 2 vary in wide limits. What are realistic values?

Check throughout the text and use Figure and Table

The authors analyze images and do not use any visual quality metrics, I prefer to see the results for, at least, one visual quality metric.

Have the filter parameters to be determined for each particular infrared scanner separately?

Author Response

(The authors gave the same response as above.)

Reviewer 3 Report

In this study, authors present a single-frame image-based non-uniformity correction technique for high resolution infrared scan images

Comments to the author:

1.      Improving English language can enhance the quality.

2.      Add more recent and latest references in support of your work.

3.      In the second row of Table 1, it should be noted that the numbers are related to PSNR.

4.      It seems better to use "proposed method" in instead of "ours" in Figures 8 and 9.

5.      In tables 3, 4, 5 and 6, it seems advisable use the word "proposed method" place of the word "ours."

Author Response

(The authors gave the same response as above.)

Round 2

Reviewer 1 Report

The authors have addressed all the revision comments.

Reviewer 2 Report

I am satisfied by the answers and corrections done. Meanwhile, I'd like to recommend the authors to read more about modern visual quality metrics.